# Intravitreal Injection Planning during COVID-19 Pandemic: A Retrospective Study of Two Tertiary University Centers in Italy

**DOI:** 10.3390/healthcare11030287

**Published:** 2023-01-17

**Authors:** Daniela Mazzuca, Giuseppe Demarinis, Marcello Della Corte, Fiorella Caputo, Antonello Caruso, Margherita Pallocci, Luigi Tonino Marsella, Filippo Tatti, Emanuele Siotto Pintor, Lorenzo Mangoni, Gabriele Piccoli, Adriano Carnevali, Sabrina Vaccaro, Vincenzo Scorcia, Enrico Peiretti, Carmelo Nobile, Nicola Gratteri, Giuseppe Giannaccare

**Affiliations:** 1Department of Surgical and Medical Sciences, University ‘Magna Græcia’ of Catanzaro, Viale Europa, 88100 Catanzaro, Italy; 2Department of Surgical Sciences, Eye Clinic, University of Cagliari, Via Ospedale 48, 09124 Cagliari, Italy; 3Department of Biomedicine and Prevention, University of Rome “Tor Vergata”, Via dell’Archiginnasio, 00133 Rome, Italy; 4Department of Ophthalmology, University ‘Magna Græcia’ of Catanzaro, Viale Europa, 88100 Catanzaro, Italy; 5Department of Health Sciences, University ‘Magna Græcia’ of Catanzaro, Viale Europa, 88100 Catanzaro, Italy; 6Department of Pharmacy, Health and Nutritional Sciences, University of Calabria, 87036 Cosenza, Italy; 7Department of Law, University ‘Magna Græcia’ of Catanzaro, Viale Europa, 88100 Catanzaro, Italy

**Keywords:** eye, intravitreal injections, ethical, COVID, medico-legal, retinal diseases, epidemiology, ophthalmology, public health

## Abstract

The COVID-19 pandemic has hampered the optimum management of retinal diseases. This study examined the impact of the pandemic on the intravitreal-injection practice in two academic centers in Italy along with the related medico-legal implications. A retrospective analysis of electronic medical records from 16 March 2020 to 14 March 2021 at the ophthalmological departments of University of Cagliari (SGD) and University Magna Græcia of Catanzaro (UMG) was conducted. The data collected between 16 March 2020 and 14 June 2020 (lockdown), 15 June 2020 and 13 September 2020 (unlock), and 14 September 2020 and 14 March 2021 (second wave) were compared with those of the same period of the previous year. Weekly data on the administered drug and the number and type of treated disease were collected and analyzed. During the lockdown, a drop of 59% at SGD (*p* < 0.00001) and 77% at UMG (*p* < 0.00001) in intravitreal injections was found. In the first year of the pandemic, the reduction in injections was approximately of 27% (*p* < 0.0008) and 38% (*p* < 0.0001) at SGD and UMG, respectively. The COVID-19-related containment measures and the health resources redistribution have led to a delay in the treatment of chronic diseases of the retina, prioritizing the undeferrable ones. The lack of management guidelines has conceived relevant ethical and medico-legal issues that need to be considered in future measures planning.

## 1. Introduction

Intravitreal injection (IVI) represents the treatment of choice in several retinal diseases and currently the most commonly performed intraocular procedure [1]. In the last decade, the availability of intravitreal therapy has expanded substantially and the number of injections performed each year has increased 100-fold [2]. The evidence shows a strong relation between the clinical outcomes and the continuity of care and intensive follow-up [3,4]; however, the protocols should take into account that patients receiving IVIs are generally old and suffer from other underlying medical conditions such as diabetes, hypertension, and cardiovascular disease [5]. To cope with the rising demand for IVIs and reduce the burden for patients, caregivers, and physicians, various approaches have been used, including different treatment protocols and therapeutic options, nurse-delivered intravitreal injections, arc sterile setting, and sustained-release drugs able to minimize the injection frequency and the cumulative number of IVIs and visits [1,6,7].

In December 2019, a cluster of cases suffering from pneumonia of unknown cause attracted global attention [8]. Several days later, a new variant of the virus was identified, which was named “severe acute respiratory syndrome coronavirus 2” (SARS-CoV-2) [9]. This is a coronavirus (positive-sense single-stranded RNA virus) transmitted by person-to-person contact or via airborne droplets or fomites [10], responsible for an infectious respiratory disease named coronavirus disease 2019 (COVID-19). After binding to epithelial cells in the respiratory tract, SARS-CoV-2 begins replication and migrates into the airways. Upon exposure to the virus, the body triggers the primary immune response [11]. In some patients, a cytokine storm syndrome may occur, causing acute respiratory distress syndrome and respiratory failure, which is considered the leading cause of death [12].

The COVID-19 pandemic has faced unprecedented challenges to healthcare services worldwide. In particular, its impact has been considerable in the ophthalmological practice [13]. The infection transmission risk, both for healthcare workers and patients, has made providing non-stop sight-saving treatments arduous and challenging. Many changes in the standard treatment protocols were adopted, and several considerations were made during the first year of the pandemic. The aim of this study was to investigate how the containment measures and the inpatient fluctuation influenced the practice patterns among retina specialists during the first year of the COVID-19 pandemic.

## 2. Materials and Methods

### 2.1. Study Design and Settings

This was a multicenter, retrospective, observational study performed in 2 ophthalmological departments in Italy: (1) San Giovanni di Dio (SGD), University of Cagliari, one of the largest hospitals in Sardinia, an island in Southern Italy; (2) Mater Domini (UMG), University Magna Græcia of Catanzaro, the largest tertiary-care ophthalmological department in Calabria, a region in the southern part of Italy. The activity of both outpatient clinics for intravitreal injections performed during the first year of the pandemic was collected and compared with the same period of the previous year. The study was approved by the local ethical committee of both hospitals (Azienda Ospedaliero-Universitaria di Cagliari, protocol code PG/2021/19405, date of approval 22/12/2021; University Magna Græcia of Catanzaro, protocol code 322-2021, date of approval 21/10/2021).

### 2.2. Data Collection

The activity of IVIs in both units from 16 March 2020 to 14 March 2021 were collected. This period was analyzed weekly for a total of 52 weeks (“Covid-year”) and was compared with the analogous period of the previous year—from 18 March 2019 to 15 March 2020 (“pre-COVID-19”). The COVID year was divided into 3 periods according to the national pandemic containment measures: (1) the first quarter, from 16 March 2020 to 14 June 2020 (“lockdown”); (2) the second quarter, from 15 June 2020 to 13 September 2020 (“unlock”); (3) the second half, from 14 September 2020 to 14 March 2021 (“second wave”). The first phases (1) and (2) consisted of 13 weeks each, while the third phase consisted of 26 weeks, according to the containment measures imposed by the Italian government [13]. Similarly, the pre-COVID year was divided into 3 equivalent periods: (1) the first quarter, from 18 March 2019 to 15 June 2019; (2) the second quarter, from 16 June 2019 to 15 September 2019; (3) the last half year, from 16 September 2019 to 15 March 2020. Study data were obtained from the Informatics and Computing Infrastructure database, which stores all information obtained from the electronic medical record (computerized patient record system) and segregated into 2 excel sheets in Microsoft Excel 2019 (Microsoft Corporation, Redmond, WA, USA). The data contained patient-specific information about diagnosis (neovascular age-related macular degeneration (wet-AMD), diabetic macular edema (diabetic ME), macular edema secondary to retinal vein occlusion (RVO ME), and choroidal neovascularization secondary to uncommon causes (other-CNV)), number of intravitreal injections, and the active substance of drug injected (anti-vascular endothelial growth factor (VEGF) therapy vs sustained-release dexamethasone intravitreal implant (DEX-I). The excel sheets with the required data were then used for analysis using the appropriate statistical software.

### 2.3. Statistical Analysis

Descriptive statistics were used to elucidate the diagnosis and drug-administered data using Microsoft Excel^®^. Continuous variables were tested with the Kolmogorov– Smirnov test for normal distribution. Continuous outcomes were described as mean (±standard deviation) and compared with Student’s *t*-test. All *p*-values are the results of two-sided tests. The percentage of use of DEX-I compared to anti-VEGF therapy for DME and RVO were evaluated using the χ^2^ test. The statistical analysis was performed using IBM SPSS Statistics 28.0 for Mac (IBM^®^, Chicago, IL, USA). Statistical significance was defined as *p* < 0.05.

## 3. Results

The difference between the diagnosis and IVI administrations in the periods is presented in Table 1 and Table 2. During the COVID year, 1732 (33.31 ± 16.14) and 1229 (23.63 ± 13.38) IVIs were performed at SGD and UMG, respectively; the difference was statistically significant compared to the pre-COVID year (*p <* 0.001) (Table 1). The most significant reduction in weekly IVIs was reported during the lockdown phase in both hospitals with a 59% drop at SGD (*p* < 0.00001) and 77% drop at UMG (*p* < 0.0001); in the other phases, the reduction was less than 20%, except for a 24% reduction at UMG during the second wave (*p* < 0.0001) (Table 1). The overall IVI trend in both hospitals is shown in Figure 1.

The most treated retinal pathology in the pre-COVID year was wet-AMD in both hospitals (weekly mean 28.08 ± 12.68 and 17.19 ± 8.17 at SGD and UMG, respectively). These numbers reduced significantly in both hospitals during the COVID year (SGD 27% *p* < 0.0002, UMG 38% *p* < 0.00001). A similar trend was observed in diabetic ME and RVO ME. The bar charts in Figure 2 illustrate the different trend in retinal diseases treated during the study period. A statistically significant reduction in SGD and UMG was observed for both diseases in the lockdown phase and in the entire pre-COVID year, while during the second wave, the reduction was significant only at UMG (Table 2).

The comparison of anti-VEGF and DEX-I for the treatment of diabetic ME and RVO ME is displayed in the Figure 3. DEX-I was significantly less used at SGD during the lockdown (15%) than in the same period of the previous year (34%, *p* < 0.00047); in the unlock phase, the percentage of DEX-I use statically increased by 12% compared to the second quarter of the previous year (*p* < 0.01), whereas in the second wave, the percentage of use was similar to the pre-COVID year (Table 3). A similar trend was observed at UMG with a statistical increase in the use of DEX-I by an amount of 15% during the unlock phase (*p* < 0.002).

## 4. Discussion with Ethical and Medico-Legal Perspectives

IVIs are the first-line treatment for patients with wet-AMD, diabetic ME, RVO ME, and other-CNV [14]. Wet-AMD is the leading cause of irreversible blindness in developed countries. In real-world practice, its management starts with three-monthly loading doses followed by one of the following protocols: (i) fixed regimen, a monthly intravitreal injection, not commonly used outside clinical trials due to the burden on the patient and healthcare personnel; (ii) pro re nata (PRN), consisting of fixed follow-up intervals and resumption of treatment if signs of disease recurrence are found; (iii) treat and extend (T&E), where a single injection is performed at each visit but the time interval and follow-up visits are planned depending on the clinical course [15,16]. These protocols are also used in the treatment of diabetic retinopathy and RVO with satisfactory results [17,18,19]. In both of the above diseases, the decrease in visual acuity (VA) is associated with macular edema, caused by the inflammation and rupture of the blood–retinal barrier [20]. Therefore, two steroid implants have been developed to treat them and to reduce the number of IVIs. The first, a dexamethasone implant (DEX-I), is useful in decreasing the number of visits and facilitating compliance in real-world practice thanks to its longer activity (≤6 months) [21]. The second, a fluocinolone acetonide implant (FAc), has been approved for the treatment of DME and was found to be released into the eye over a 36-month period [22]. Nonetheless, anti-VEGF treatment according to the PRN protocol remains the best choice for the treatment of the other-CNV [23]. The sudden spread of SARS-CoV-2 and the nationwide lockdown measures have led to a dramatic decrease in patients seeking emergency care or attending scheduled hospital appointments [24]. In this paper, we report our experience with the administration of IVI in the first year of COVID-19, in accordance with the restriction measures imposed by the National Government and the medical offices of Italian hospitals to flatten the curve of viral spread [13,25].

During the lockdown period, we observed a significant decrease in IVI practice of 59% and 77% at SGD and UMG, respectively (Table 1). Similar results have been reported in several hospitals both in Italy and worldwide [26,27]. The IVI decrease showed a different impact in both hospitals for each disease (Table 2). We found a significant reduction for the three main indications: wet-AMD, diabetic ME, and RVO ME (Figure 1a,b). However, while the decline tended to be similar for each of the indications at SGD, significant differences were observed at UMG, with the decline being more pronounced for diabetic ME (83%) (Table 2). This scenario may be explained by the containment measures imposed by the government and medical office, and by the fragile and elderly patients’ reluctance to undergo hospital treatment for their comorbidities [28,29]. The postponement of both deferrable surgical and clinical activities raises the issue of identifying which diseases require more urgent treatment. Prior to the era of COVID, no priority criteria were set in the guidelines for disease management [2]. As the pandemic started, several ophthalmological societies sought to provide guidance to retina specialists in this uncharted territory [30]. In March 2020, the American Academy of Ophthalmology (AAO) specified that the decision between “non-urgent” and “urgent” medical conditions depends on the physician’s assessment, and the American Society of Retina Specialists (ASRS) defined all IVIs as essential [31,32]. In Europe, the Société Française d’Ophtalmologie (SFO) proposed two different approaches depending on the pathology: in wet-AMD, consultations were to be cancelled and injections were to be followed at a fixed interval and, if possible, bilaterally on the same day; in diabetic ME and RVO ME, deferral could be considered in most cases [33]. Similarly, the Royal College of Ophthalmologists (RCOphth) has issued a “medical retinal management plan during COVID-19” based on the disease: in wet-AMD, the loading dose had to be maintained, while subsequent treatments had to be administered at 8-week intervals; in the case of diabetic ME and RVO ME, the injection could be deferred [34]. These different treatment strategies, coupled with the reluctance of some patients to undergo IVIs and the decrease in outpatient activity due to the dramatic increase in COVID patients [13], resulted in a common and consistent drop in injections in both clinics during the lockdown (Figure 2), a statistically significant difference in terms of diseases treated (Table 2), and the inability to follow normal treatment protocols. Similarly, Arruabarrena et al. conducted a study on the impact of the pandemic in wet-AMD patients in the ophthalmological departments of six centers of Europe and reported a reduction in the mean number of injections during the lockdown followed by a significant reduction in the mean best-corrected visual acuity [35].

At the same time, in “Guidance for anti-VEGF intravitreal injections during the COVID-19 pandemic”, Korobelnik et al. emphasized the importance of maintaining the schedule for the loading phase; in diabetic ME/RVO ME patients over DEX-I, re-implantation would have been considered only in cases of good response and no history of adverse effects [36]. These suggestions had led to different trends in the use of DEX-I during the lockdown, furtherly supported by the lower risk of irreversible vision loss in patients with diabetic ME and RVO ME in the short term [37,38]. In the lockdown, we observed a significant decrease in the percentage of use of DEX-I compared to anti-VEGF at SGD (Figure 3), while it was slightly increased in the UMG group, in line with the findings of the other centers, both in Italy and worldwide [27,30]. Meanwhile, for FAc, real-life data showed a higher rate of side effects related to the need of surgery for elevated IOP and cataract development than DEX-I [39]. For the above reasons and the unavailability of real-world data during the COVID-19 pandemic, in both hospitals, DEX-I has been selected as the only second-choice therapeutic option.

During the second phase, the “unlock phase”, a reduction in the IVI activity was observed (Figure 2), albeit to a lesser extent than during the first months of the pandemic. Interestingly, both clinics displayed an interesting increase of 10–15% in DEX-I use vs anti-VEGF compared to the pre-COVID year (Figure 3). This trend change may be explained as follows: Firstly, DEX-I’s properties give the possibility to control the disease for a long period of time; thereby, it may reduce the number of visits [21]. Secondly, the switch to DEX-I from anti-VEGF could grant some visual and anatomical benefits in DME patients, the principal disease treatable with this drug, as demonstrated in real-world settings [21,22,40]. Thirdly, the course of events since the end of the unlocking phase had indicated that there would have been a new surge of infection and a new reduction in medical activities. Therefore, this treatment was likely to partially reduce future workload and prevent a rebound effect, as predicted by Borrelli et al. [13,41,42]. Recent research, which analyzed French National Health Data, confirmed the rebound effect and showed a relatively marked decrease in the dispensing of anti-VEGF IVIs during the lockdown, which was not fully compensated for during the first 4 weeks after unlocking [43].

As expected, the number of contagions gradually increased during the second wave and new containment measures were adopted [13]. In both clinics, there was a renewed decline in IVI activity, but it was less severe than during the lockdown phase (Figure 2). However, despite the infections increasing, the IVI activity has resumed (Figure 2). This may be due to the new patient- and ophthalmologist-protective measures developed during the lockdown, as well as the awareness of the dangers of delaying sight-saving treatment [44]. Despite the fact that a slow recovery has followed the lockdown, the backlog of performed IVIs is still high at both clinics throughout the COVID year: both SGD and UMG displayed a statistically significant reduction in the number of IVIs of 27% and 38%, respectively (Table 1).

From an ethical point of view, the COVID pandemic poses a major challenge to the field of ophthalmology: on the one hand, the need to protect people’s lives from a potentially fatal risk of infection [45], and on the other, the need to safeguard sight and to prevent a possible progression to blindness. Therefore, we believe that an ethically relevant issue should be analyzed in a medico-legal frame, especially with regard to the necessary treatments and follow-up visits for chronic diseases, the delay of which could lead to irrecoverable visual damage. During the worst periods of the COVID pandemic and sanitary emergency, ophthalmologists around the world tried to cope with the impossibility of visiting patients on site through the implementation of telemedicine programs and activities [46]. The unexplored field of pandemic events is still difficult to face, especially with regard to the deferral of treatment of chronic illnesses, which were defined as “obviously necessary, but not necessarily urgent”. In the field of medical retina, the most important scientific societies have drawn up various guidelines and clinical suggestions with different and divergent solutions, making the issue arduous and still open. Even before the pandemic, several studies described that, in real-world practice, macular disease patients reported less visual-acuity gain due to undertreatment than in clinical trials [47,48]. A strong correlation between delaying treatment and a worse anatomical and visual outcome, even in the short term, has been widely reported [49]. In addition, it is becoming increasingly evident that early diagnosis and regular IVI are necessary to achieve an improvement in visual acuity [50]. After the pandemic began, this trend worsened considerably and, as shown by our study and reported by several authors, numerous IVIs were deferred or eliminated with dramatic anatomical and visual outcomes [51,52].

Unfortunately, these terrible results not only alert us to a possible increase in the number of visually impaired individuals in the coming years, but also draw attention to the resulting increase in future healthcare costs [53]. Indeed, worsening visual acuity directly results in an increase in disability with the associated negative effects, mainly on the patient’s quality of life, and secondly on the Italian economy and healthcare systems, a still poorly considered field. Indeed, retinal diseases’ chronicity is estimated to be responsible for significant economic and social costs. In particular, they absorb a consistent part of economic resources and this phenomenon may grow in the future in consideration of expected population ageing [47].

However, the study also has its limitations. We examined the impact of the first year of the pandemic on IVI clinic volume and indications for injection, but a major limitation could be data collection bias due to the retrospective nature of the study, and the statistical significance of the results could be related to the random distribution within the study population. At the same time, the correlations with the anatomical and functional data were not evaluated due to the complex COVID-19 situation, which made patient access to hospitals very problematic.

## 5. Conclusions

The relevant consequences on intravitreal-injection activity were reported in the early months of the pandemic. Increased awareness of sight-saving-treatment administration and a resilience program in IVI planning partially reduced the gap that developed during the lockdown phase. Nevertheless, the backlog remains significant. Given the importance of these treatments and the risk of increasing the number of visually impaired people, forecasting new future trends may help healthcare systems to plan resources and secure financing to be able to accommodate the expected demand.

## Figures and Tables

**Figure 1 healthcare-11-00287-f001:**
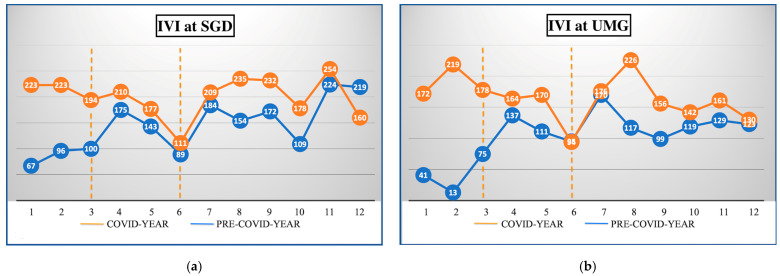
The monthly number of intravitreal injections (IVIs) performed during the COVID year (blue) compared to pre-COVID year (orange) at San Giovanni di Dio (SGD) (**a**) and University Magna Græcia of Catanzaro (UMG) (**b**). The dotted orange line in each figure divided the period studied into three phases: lockdown, unlock, and second wave from left to right, respectively.

**Figure 2 healthcare-11-00287-f002:**
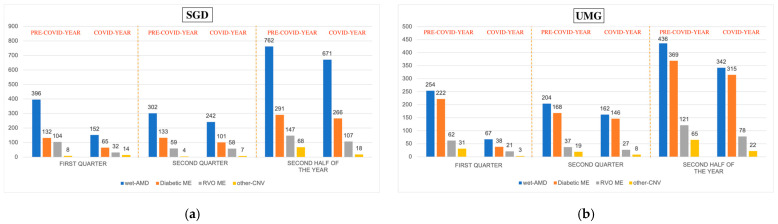
The change in the number of intravitreal injections divided into 4 categories: wet-AMD (blue), diabetic ME (orange), RVO ME (grey), and other-CNV (yellow) performed at San Giovanni di Dio (SGD), University of Cagliari (**a**) and Mater Domini, University Magna Græcia of Catanzaro (UMG) (**b**), in each study period.

**Figure 3 healthcare-11-00287-f003:**
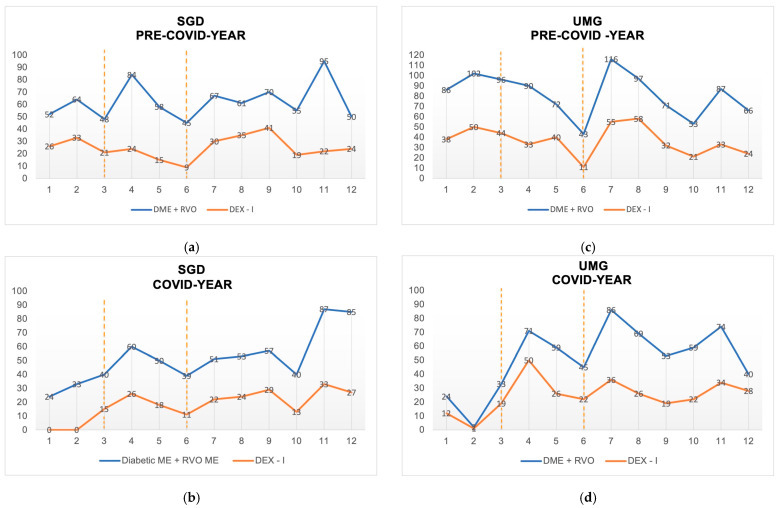
The monthly number of diabetic macular edema (diabetic ME) and macular edema secondary to retinal vein occlusion (RVO ME) treated (blue) and the sustained-release dexamethasone intravitreal implant (DEX-I, orange): (**a**) pre-COVID year and (**b**) COVID year at San Giovanni di Dio (SGD), (**c**) pre-COVID year, and (**d**) COVID year at University Magna Græcia of Catanzaro (UMG). The dotted orange line in each figure divides the period studied into three phases: lockdown, unlock, and second wave from left to right, respectively.

**Table 1 healthcare-11-00287-t001:** Number of intravitreal-injection procedures performed at San Giovanni di Dio (SGD), University of Cagliari and Mater Domini, University Magna Græcia of Catanzaro (UMG). Note: statistically significant values are in italics.

	SGD		UMG	
	No of IVI (Weekly Mean ± SD)	∆ (*p*)	No of IVI (Weekly Mean ± SD)	∆ (*p*)
FIRST QUARTER	640 (49.23 ± 17.80)	*−58.91%* *−0.00001*	569 (43.77 ± 2.71)	*−77.31%* *−0.0001*
LOCKDOWN	263 (20.23 ± 6.90)		129 (9.92 ± 12.86)	
SECOND QUARTER	498 (38.30 ± 26.17)	−16.60% −0.47	428 (32.92 ± 20.79)	−19.86% −0.35
UNLOCK	407 (31.31 ± 17.79)		343 (26.38 ± 13.52)	
SECOND HALF OF THE YEAR	1268 (48.77 ± 19.88)	−15.24% −0.12	991 (38.12 ± 17.18)	*−23.61%* *−0.0001*
SECOND WAVE	1062 (40.85 ± 14.36)		757 (29.12 ± 8.08)	
PRE-COVID YEAR	2406 (46.27 ± 21.24)	*−27.26%* *−0.0008*	1988 (38.23 ± 16.22)	*−38.18%* *−0.0001*
COVID YEAR	1732 (33.31 ± 16.14)		1229 (23.63 ± 13.38)	

**Table 2 healthcare-11-00287-t002:** Comparison of different retinal diseases treated at San Giovanni di Dio (SGD), University of Cagliari and Mater Domini, University Magna Græcia of Catanzaro (UMG). Note: statistically significant values are in italics.

SGD								
	Wet-AMD (Weekly Mean ± SD)	∆ *(p)*	Diabetic ME (Weekly Mean ± SD)	∆ *(p)*	RVO ME (Weekly Mean ± SD)	∆ *(p)*	other-CNV (Weekly Mean ± SD)	∆ *(p)*
FIRST QUARTER	396 (30.46 ± 10.47)	*−62% (0.00007)*	132 (10.15 ± 5.03)	*−51%* *(0.01)*	104 (8.00 ± 4.60)	*−69% (0.006)*	8 (0.62 ± 0.77)	+75% (0.11)
LOCKDOWN	152 (11.69 ± 4.57)		65 (5.00 ± 2.31)		32 (2.46± 1.13)		14 (1.08 ± 0.86)	
SECOND QUARTER	302 (23.25 ± 15.61)	−20% (0.31)	133 (10.23 ± 7.88)	−29% (0.28)	59 (4.54 ± 4.05)	−8% (0.87)	4 (0.31 ± 0.63)	+75% (0.31)
UNLOCK	242 (18.62 ± 11.91)		95 (7.31 ± 5.01)		54 (4.15 ± 3.02)		7 (0.54 ± 0.52)	
SECOND HALF OF THE YEAR	762 (29.31 ± 11.90)	*−12% (0.09)*	291 (11.19 ± 5.48)	−9% (0.40)	147 (5.65 ± 4.01)	−27% (0.10)	68 (2.62 ± 2.17)	*−74% (0.0001)*
SECOND WAVE	671 (25.91 ± 9.02)		266 (10.23 ± 5.18)		107 (4.12 ± 2.58)		18 (0.69 ± 1.01)	
PRE-COVID YEAR	1460 (28.08 ± 12.68)	*−27% (0.0002)*	556 (10.69 ± 5.96)	*−23%* *(0.01)*	310 (1.54 ± 1.93)	*−38% (0.001)*	80 (1.54 ± 1.93)	*−51%* *(0.009)*
COVID YEAR	1065 (20.48 ± 10.66)		426 (8.19 ± 5.02)		193 (0.75 ± 0.88)		39 (0.75 ± 0.88)	
**UMG**								
	wet-AMD (weekly mean ± SD)	∆ *(p)*	Diabetic ME (weekly mean ± SD)	∆ *(p)*	RVO ME (weekly mean ± SD)	∆ *(p)*	other-CNV (weekly mean ± SD)	∆ *(p)*
FIRST QUARTER	254 (19.54 ± 2.44)	*−74% (0.00001)*	222 (17.08 ± 2.60)	*−83% (0.000001)*	62 (4.77 ± 2.01)	*−66% (0.0007)*	31 (2.38 ± 1.50)	*−90% (0.00001)*
LOCKDOWN	67 (5.15 ± 5.43)		38 (2.92 ± 4.82)		21 (1.62 ± 2.75)		3 (0.23 ± 0.60)	
SECOND QUARTER	204 (15.69 ± 10.86)	−21% (0.31)	168 (12.92 ± 8.75)	−12% (0.45)	37 (2.85 ± 2.27)	−27% (0.33)	19 (1.46 ± 1.90)	−58% (0.33)
UNLOCK	162 (12.46 ± 6.65)		146 (11.30 ± 6.64)		27 (2.08 ± 1.89)		8 (0.62 ± 0.65)	
SECOND HALF OF THE YEAR	436 (16.77 ± 8.55)	*−25% (0.006)*	369 (14.19 ± 8.00)	*−17%* *(0.05)*	121 (4.65 ± 2.58)	*−37% (0.01)*	65 (2.50 ± 2.08)	*−72% (0.0002)*
SECOND WAVE	342 (12.54 ± 5.12)		315 (11.73 ± 4.47)		78 (2.92 ± 2.23)		22 (0.69 ± 0.97)	
PRE-COVID YEAR	894 (17.19 ± 8.17)	*−38% (0.00001)*	759 (14.60 ± 7.30)	*−35% (0.0001)*	220 (4.23 ± 2.46)	*−44% (0.0001)*	115 (2.21 ± 1.92)	*−75% (0.0000003)*
COVID YEAR	555 (10.67 ± 6.37)		491 (9.44 ± 6.33)		124 (2.38 ± 2.32)		29 (0.56 ± 0.83)	

**Table 3 healthcare-11-00287-t003:** Comparison of anti-VEGF and sustained-release dexamethasone intravitreal implant (DEX-I) treatment for diabetic macular edema (diabetic ME) and macular edema secondary to retinal vein occlusion (RVO ME). Note: statistically significant values are in italics.

	Diabetic ME and RVO ME							
	SGD				UMG			
	ANTI-VEGF	DEX-I	∆ DEX-I	*p*	ANTI-VEGF	DEX-I	∆ DEX-I	*p*
FIRST QUARTER	157	80	33.76%	*0.00047*	152	132	46.48%	0.27
LOCKDOWN	82	15	15.46%		27	32	54.24%	
SECOND QUARTER	144	48	25.00%	*0.01*	121	84	40.98%	*0.002*
UNLOCK	94	55	36.91%		75	98	56.65%	
SECOND HALF OF THE YEAR	267	171	39.04%	0.85	267	223	45.51%	0.29
SECOND WAVE	225	148	39.68%		228	165	41.98%	
PRE-COVID YEAR	567	299	34.53%	0.78	540	439	44.84%	0.22
COVID YEAR	401	218	35.22%		320	295	47.97%	

## Data Availability

Not applicable.

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
