# Peer review of "Intravitreal Injection Planning during COVID-19 Pandemic: A Retrospective Study of Two Tertiary University Centers in Italy"

_healthcare, 2023, doi:10.3390/healthcare11030287_

Round 1

Reviewer 1 Report

This article addresses a topic of considerable importance and current relevance, concerning some important issues related to intravitreal injections planning during COVID-19 pandemic.

Here are some comments and/or suggestions:

Page 2, line 48: It would be more accurate to say: the most commen performed intraocular procedure

Page 3, line 98: Would the authors please explain why they did not include patients with uveitic macular oedema?

Results: It would be really useful if authors could report the impact of this reduction in the number of IVIs on the antomic and functional outcome of the patients (visual acuity and macular thickness).

Page 5, line 158: Fixed protocol must also be mentioned.

Page 7, line 223-232: The authors discussed the changes in the use of DEX-I during the lockdown, but it would also be relevant to refer to other long-acting agents such as fluocinolonacetonid and brolucizumab, which was launched in 2020. Was there a tendency towards these long-acting agents due to pandemic?

The impact of the COVID-19 lockdown period on inpatient and outpatient volumes in a tertiary referral centre in other European countries has also been reported in similar publications, please report and discuss.

Author Response

Reviewers’ comments are showed in normal text, authors’ replies in bold.

Reviewer #1

Comments and Suggestions for Authors

This article addresses a topic of considerable importance and current relevance, concerning some important issues related to intravitreal injections planning during COVID-19 pandemic.

Here are some comments and/or suggestions:

Page 2, line 48: It would be more accurate to say: the most common performed intraocular procedure

Thank you for your suggestion. Correction has been made in the revised manuscript.

Page 3, line 98: Would the authors please explain why they did not include patients with uveitic macular oedema?

Thank you for your comment. We decided to include in our study patients affected by the most common retinal diseases that are treated with intravitreal injection of drug (age-related macular degeneration, diabetic macular oedema, retinal vein occlusion and other types of CNV).

Results: It would be really useful if authors could report the impact of this reduction in the number of IVIs on the anatomic and functional outcome of the patients (visual acuity and macular thickness).

Thank you for pointing out this issue. However, during the acute phase of COVID-19 pandemic the accesses of patients to the Hospital were limited and the collection of anatomical and functional data was not performed according to a regular timeline. We added this point in the limitation section of the paper.

Page 5, line 158: Fixed protocol must also be mentioned.

Correction has been made in the revised manuscript (page 7, line 234-235).

Page 7, line 223-232: The authors discussed the changes in the use of DEX-I during the lockdown, but it would also be relevant to refer to other long-acting agents such as fluocinolonacetonid and brolucizumab, which was launched in 2020. Was there a tendency towards these long-acting agents due to pandemic?

We really appreciate your comment. Following your suggestion, we added a section about fluocinolone acetonide in the discussion (page 7, line 246-248 and page 8 line 635-639). Concerning brolucizumab, it was unfortunately not possible to describe our experience because it became available in our 2 hospitals in 2021 with an indication of use limited to a clinical trial.

The impact of the COVID-19 lockdown period on inpatient and outpatient volumes in a tertiary referral centre in other European countries has also been reported in similar publications, please report and discuss.

Thank you for pointing this out. We reported the experience of tertiary referral centre in other European countries (page 8, line 621-625 and page 8, line 651-654)

Reviewer 2 Report

Dear Authors,

Manuscript ID: healthcare-2091977, "Medico-Legal and Ethical Issues in Intravitreal Injections Planning during COVID-19 Pandemic: the Experience of two Tertiary University Centers in Italy," is fairly interesting. My primary concern with this study is that it is primarily descriptive, lacks supporting evidence, and provides inadequate evidence to support medicolegal and ethical issues on this study. Reading this, I was left wondering what exactly the point of this retrospective analysis was supposed to be. The author could have better served the reader by establishing the relationship between containment measures and blindness, as well as COVID-19 infection and outpatient activities during IVI in their study population. Please address this in the revised version.

Some of my concerns are below.

§  The manuscript's data do not support the medicolegal and ethical stance. Therefore, please revise the title in accordance with the retrospective analysis.

§  In 2nd paragraph of the introduction, briefly discuss COVID-19 and SARS-SoV-2 and the immune response against them. Some useful references PMID: 34508515, PMID: 33024307.

§  Please correct to make COVID-19 on line 66

§  Please write the full form of IVT in table 1 or make it the same as IVI.

§  Please correct to make (P<0.001) on line 116.

§  Figures 1, 2, and 3 should be in the results section, briefly explaining what they represent. Also, figures are barely visible, so separate "a" and "b" to keep high-quality, readable images.

§  The discussion must be short and to the point for this limited observational study. Please don't overstate those that aren't supported by good evidence.

§  Perhaps Table 4 is redundant for an observational study of this nature.

§  The statistical significance shown in the results may be linked to random distribution within the study population. Therefore, please highlight these points as a study limitation.

§  It is unclear what "Dramatic consequences on intravitreal injection activity were reported" refers to in line 333. Please correct it.

§  Please provide concluding remarks based on the data rather than simply babbling in the conclusion.

Author Response

Reviewers’ comments are showed in normal text, authors’ replies in bold.

Reviewer #2

Dear Authors,

Manuscript ID: healthcare-2091977, "Medico-Legal and Ethical Issues in Intravitreal Injections Planning during COVID-19 Pandemic: the Experience of two Tertiary University Centers in Italy," is fairly interesting. My primary concern with this study is that it is primarily descriptive, lacks supporting evidence, and provides inadequate evidence to support medicolegal and ethical issues on this study. Reading this, I was left wondering what exactly the point of this retrospective analysis was supposed to be. The author could have better served the reader by establishing the relationship between containment measures and blindness, as well as COVID-19 infection and outpatient activities during IVI in their study population. Please address this in the revised version.

Some of my concerns are below.

  • The manuscript's data do not support the medicolegal and ethical stance. Therefore, please revise the title in accordance with the retrospective analysis.

Thank you for your suggestion. Correction has been made in the revised manuscript.

  • In 2nd paragraph of the introduction, briefly discuss COVID-19 and SARS-SoV-2 and the immune response against them. Some useful references PMID: 34508515, PMID: 33024307.

Thank you for your suggestion. A brief discussion about COVID-19 and SARS-CoV-2 has been added in the introduction (page 2, line 75-86)

  • Please correct to make COVID-19 on line 66

Correction has been made in the revised manuscript.

  • Please write the full form of IVT in table 1 or make it the same as IVI.

Correction has been made in the revised manuscript.

  • Please correct to make (P<0.001) on line 116.

Correction has been made in the revised manuscript.

  • Figures 1, 2, and 3 should be in the results section, briefly explaining what they represent. Also, figures are barely visible, so separate "a" and "b" to keep high-quality, readable images.

Thanks for your suggestion, Figure 1,2 and 3 has been separated and moved in the result section

  • The discussion must be short and to the point for this limited observational study. Please don't overstate those that aren't supported by good evidence.

Thank you for your consideration. We deleted the sentences that are not supported by enough evidence and we have reduced the length of the discussion section.

  • Perhaps Table 4 is redundant for an observational study of this nature.

Correction has been made in the revised manuscript.

  • The statistical significance shown in the results may be linked to random distribution within the study population. Therefore, please highlight these points as a study limitation.

Thank you for your comment, following your suggestion, we added a section with the limitation of the study (page 9, line 1002-1006).

  • It is unclear what "Dramatic consequences on intravitreal injection activity were reported" refers to in line 333. Please correct it.

Correction has been made in the revised manuscript.

  • Please provide concluding remarks based on the data rather than simply babbling in the conclusion.

Thank you for your suggestion. The conclusion was changed in order to be focused more on the topic of the paper.

Round 2

Reviewer 1 Report

The authors were able to answer most of the raised questions.

Reviewer 2 Report

Dear Authors,

The revised version is  well-addressed, reasonably convincing and transparent.  Now I can endorse the draft for publication.

Thank you